# RAG©: Towards Copyright Protection for Knowledge Bases of Retrieval-augmented Language Models

## Abstract

Large language models (LLMs) are increasingly integrated into real-world applications through retrieval-augmented generation (RAG) mechanisms to supplement their responses with up-to-date and domain-specific knowledge. However, the valuable and often proprietary nature of the knowledge bases used in RAG introduces the risk of unauthorized usage by adversaries. Existing methods that can be generalized as watermarking techniques to protect these knowledge bases typically involve backdoor or poisoning attacks, which introduce harmful behaviors (*e.g.*, generating incorrect outputs for verification), thereby compromising the LLM's reliability. To address these challenges, we propose RAG© for harmless copyright protection of knowledge bases. Instead of manipulating the final output, RAG© implants distinct verification behaviors in the space of chain-of-thought (CoT) reasoning, maintaining the correctness of the final answer. Our approach involves three main stages: (1) **Generating CoTs**: For each verification question, we generate two CoTs, including a target CoT for building watermark behaviors; (2) **Optimizing Watermark Phrases and Target CoTs**: We optimize them to minimize retrieval errors under the black-box setting of suspicious LLM, ensuring that the watermarked verification queries activate the target CoTs without being activated in non-watermarked ones; (3) **Ownership Verification**: We exploit a pairwise Wilcoxon test to statistically verify whether a suspicious LLM is augmented with the protected knowledge base by comparing its responses to watermarked and benign verification queries. Our experiments on diverse benchmarks demonstrate that RAG© effectively protects knowledge bases against unauthorized usage while preserving the integrity and performance of the RAG.

## 1 Introduction

Large language models (LLMs), such as GPT-4 (Achiam et al., 2023), LLaVa (Liu et al., 2024), and PaLM (Anil et al., 2023), have been widely deployed in many real-world applications, including intelligent assistant (Dong et al., 2023), ChatBot (Zheng et al., 2023), and finance (Dowling & Lucey, 2023). Despite their success in exceptional generative capabilities, they also suffer from lacking up-to-date knowledge as they are pre-trained on past data (Wu et al., 2024); they could also lack knowledge on specific domains (*e.g.*, medical domain (Xiong et al., 2024)), restricting the real-world deployment of LLMs in applications like healthcare (Zakka et al., 2024).

To address the above limitations, *retrieval-augmented generation (RAG)* is proposed to augment an LLM with external knowledge retrieved from given knowledge databases. Its main idea is to combine the strengths of retrieval-based and generative models to produce more accurate and contextually relevant outputs. In general, RAG contains three main components: *LLMs*, *retriever*, and *knowledge base*. Specifically, LLMs and the retriever are both machine learning models pre-trained with existing data for generating answers and knowledge retrieval. Knowledge bases contain a large number of texts collected from various domains or the Internet to provide domain-specific expertise and up-to-date information for LLMs. In particular, these knowledge bases, especially those from mission-critical domains (*e.g.*, finance (Zhang et al., 2023) and healthcare Zakka et al. (2024)), usually contain a large amount of valuable or even exclusive data. It leads to great incentives for adversaries to 'steal' or 'misuse' these knowledge bases for enhancing their deployed LLMs service

Figure 1: The workflow of copyright protection for RAG's knowledge base with backdoor-based watermark and our RAG$^{©}$. Both backdoor-based watermarks and RAG$^{©}$ implant owner-specified watermarks into specific verification questions to activate distinctive behaviors for LLMs augmented with the protected knowledge base. However, backdoor-based watermarks are harmful, leading to incorrect answers or decisions when watermark phrases appear. In contrast, our RAG$^{©}$ implants distinctive behaviors in the space of chain-of-thought instead of directly in the final results while maintaining the correctness of final answers and decisions.

without authorization (Anderson et al., 2024). In this paper, we explore the copyright protection of knowledge bases used for RAG by detecting potential misuse.

To the best of our knowledge, no existing work has been proposed to protect the copyright for RAG's knowledge bases. Arguably, one of the most straightforward or even the only potential solutions is to formulate this copyright protection problem as an ownership verification: defenders evaluate whether a third-party suspicious LLM is augmented with their RAG knowledge base under the *black-box* and *text-only* setting, where the defender can only query the suspicious LLM with prompts and get the corresponding content through its API without accessing its parameters, configurations (*e.g.*, model structure), and intermediate results (*e.g.*, token probability). To achieve this, similar to existing dataset ownership verification (Li et al., 2020; 2022; Xu et al., 2023; Yao et al., 2024), the owners of knowledge base should first watermark it via poisoning or backdoor attacks against RAG (Chen et al., 2024; Xiang et al., 2024; Zou et al., 2024) before storing and distributing it so that all LLMs augmented with it will have some distinctive prediction behaviors. Unfortunately, these methods inevitably introduce new security risks to the deployed machine learning systems as these distinctive behaviors are generating incorrect answers or decisions on particular verification prompts. This 'harmful' nature will hinder their applications in practice (Li et al., 2022; Yao et al., 2024; Shao et al., 2025). As such, an intriguing and important question arises here: *Can we design harmless copyright protection for RAG's knowledge base?*

The answer to the above question is positive! We argue that their harmful nature is inevitable since they directly implant distinctive behaviors simply and directly in the final results. As such, they have to make the results of verification samples/prompts incorrect or anomalous to distinguish them from normal ones. Motivated by this understanding, we propose to implant these behaviors in another space, particularly the space of chain-of-thought (CoT, *i.e.*, lines of reasons). CoT is a fundamental step of LLM reasoning for its results, containing sufficient information. In general, our method (dubbed 'RAG$^{©}$') first selects a few questions (dubbed 'verification questions') and generates two different CoTs, including the target CoT and the non-target CoT (via LLMs or human experts) with the correct answer for each question, as shown in Figure 1. RAG$^{©}$ will then watermark the knowledge base based on these CoTs, leading to all 'bad RAGs' (*i.e.*, LLMs augmented with our knowledge base) answer the watermarked verification questions based on their corresponding target CoTs while their answers generated by benign LLMs are based on non-target CoTs.

Our RAG$^{©}$ consists of three main stages: **(1)** generating CoTs, **(2)** optimizing watermark phrases and target CoTs, and **(3)** ownership verification. The first stage generates two CoTs for each verification question following the above approaches; In the second stage, we first prove that the upper bound of the retrieval error rate of the target CoT is related to the similarity between the verification question containing the watermark phrase and other instances within the knowledge base without the watermark on the hidden space. Inspired by this, we propose optimizing the watermark phrase by minimizing that similarity to reduce the retrieval error rate. We design two methods, including optimization-based and LLM-based ones, to optimize the watermark phrase for each verification question. The former requires a surrogate (pre-trained) retriever to optimize the watermark directly. Inspired by (Chen et al., 2024), we also incorporate a linguistic-related loss guided by an LLM, to ensure the naturalness and fluency of the sentences after adding the watermark phrase, thereby

improving the stealthiness of watermarked verification questions used to query the suspicious LLM. The latter directly exploits LLMs to generate a phrase containing rare words that do not affect the original sentence meaning and do not influence the meaning of the original sentence, such as the watermark phrase. The optimization-based works better, but the LLM-based approach is more efficient and convenient. Besides, we also further exploit the LLM-based approach to efficiently and effectively optimize target CoTs. Intuitively, with the rare words introduced by this method, the distribution of optimized target CoTs in the embedding space will shift from the distribution of benign ones, making it more difficult to be retrieved by questions without watermarks; In the third stage, RAG$^{©}$ examines whether the suspicious LLM has been augmented with the protected knowledge base via pairwise Wilcoxon test (Schmetterer, 2012), based on the judgment of advanced LLMs (*e.g.*, GPT-4) on whether the answers of the suspicious LLM on watermarked and benign verification questions contain the information of their corresponding target CoT.

In conclusion, the main contributions of this paper are four-fold: **(1)** We explore the copyright protection of knowledge bases used for RAG and formulate this problem as an ownership verification under the black-box setting. **(2)** We reveal the harmful nature of extending existing backdoors or poisoning attacks against LLMs to watermark the knowledge base used for ownership verification. **(3)** We propose a simple yet effective harmless copyright protection of knowledge bases by implanting distinctive behaviors in the space of chain-of-thought instead of directly in the final results, and provide its theoretical foundations. **(4)** We conduct extensive experiments on benchmark datasets, verifying the effectiveness of our RAG$^{©}$ and its resistance to potential adaptive methods.

## 2  BACKGROUND AND RELATED WORK

**Retrieval-Augmented Generation (RAG).** Retrieval-augmented generation (RAG) is a technique designed to enhance the capabilities of large language models (LLMs) by integrating external knowledge sources (*i.e.*, knowledge bases) (Lewis et al., 2020). Unlike traditional LLMs, which generate responses solely based on the knowledge encoded during pre-training, RAG combines both retrieval and generation mechanisms to produce more accurate, contextually relevant, and up-to-date outputs. Existing RAG systems implemented dual encoders to map queries and texts within the knowledge base into the embedding space and retrieve candidate texts that produce high similarity values with the given query. Recent works were proposed to improve the effectiveness of retrieval models by implementing different encoder architectures (Nogueira & Cho, 2019; Humeau et al., 2019; Khattab et al., 2021), searching algorithms (Xiong et al., 2021b), embedding capacity (Günther et al., 2023), max tokens (Muennighoff et al., 2022), *etc*. In general, the knowledge base plays a critical role in the effectiveness of the RAG, containing valuable and often proprietary content. They are valuable intellectual property of their owners and their copyright deserves to be protected.

**Poisoning and Backdoor Attacks against RAG Systems.** Recently, there are also a few pioneering works exploring data-centric threats in RAG systems (Zou et al., 2024; Xiang et al., 2024; Chen et al., 2024; Cheng et al., 2024). Specifically, PoisonedRAG (Zou et al., 2024) proposed the first data poisoning attack against RAG by injecting several malicious and wrong answers into the knowledge base for each pre-defined query. The adversaries could lead the compromised RAG to generate targeted wrong answers with these pre-defined queries. TrojanAgent (Cheng et al., 2024) proposed a backdoor attack by compromising its retriever; thus, leveraging queries attaching with adversary-specified optimized trigger patterns could activate the malicious behavior embedded in its compromised retriever. Most recently, AgentPoison (Chen et al., 2024) proposed the backdoor attack against RAG by injecting optimized malicious target texts (decisions) into the external knowledge base. AgentPoison also proposed an optimization framework to optimize a stealthy and effective trigger pattern for increasing the probability of retriever retrieving the hidden malicious target texts. These methods all seriously undermine the integrity of RAG systems.

**Dataset Ownership Verification.** Dataset ownership verification (DOV) aims to verify whether a suspicious model is trained on the protected dataset (Li et al., 2022; 2023; Guo et al., 2023; Tang et al., 2023; Yao et al., 2024). To the best of our knowledge, this is currently the only feasible method to protect the copyright of public datasets in a retrospective manner. Specifically, DOV intends to introduce specific prediction behaviors (towards verification samples) in models trained on the protected dataset while preserving their performance on benign testing samples, by solely watermarking the dataset before releasing it. Dataset owners can verify ownership by examining whether

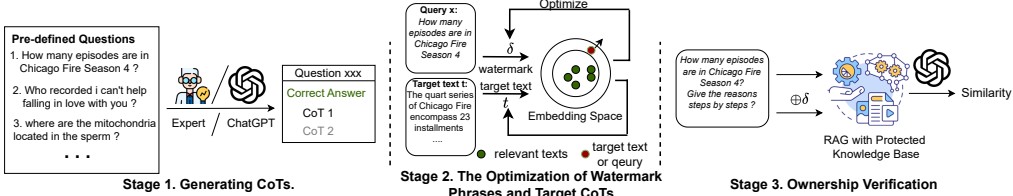

Figure 2: RAG$^{©}$ contains three main stages. In the first stage, RAG$^{©}$ requires human experts or an advanced LLM to generate the correct answer along with two distinctive CoTs for each defender-specified verification question. In the second stage, RAG$^{©}$ optimizes the watermark phrase and its corresponding target CoT texts for each verification question, aiming to cause the watermarked question and target CoT far away from the texts related to the question in the embedding space of the target retrieval model. In the third stage, RAG$^{©}$ verifies the copyright by examining whether a given suspicious LLM can generate the target CoTs for pre-defined questions. We leverage the SOTA LLM (*i.e.*, GPT-4) to help us with the investigation at this stage.

the suspicious model has dataset-specified distinctive behaviors. Previous DOV methods (Li et al., 2022; 2023; Tang et al., 2023) exploited either backdoor attacks or others (Guo et al., 2023) to watermark the original (unprotected) benign dataset or prompts. For example, backdoor-based DOV adopted poisoned-/clean-label backdoor attacks to watermark the protected dataset. Regarding the harmless copyright protection for dataset or prompt, Guo *et al.* (Guo et al., 2023) and (Yao et al., 2024) proposed harmless watermark techniques for image classification and instruction fine-tuning applications, where the watermark samples are not allowed to cause malicious behavior (*e.g.*, misclassification) for verification purposes. CPR (Golatkar et al., 2024) proposed copyright-protected RAG to provide copyright protection guarantees in a mixed-private setting for diffusion models. CPR focused on addressing privacy leakage issues in the generation procedure of diffusion models. However, the copyright protection technique for the knowledge base of RAG remains blank.

## 3 METHODOLOGY

### 3.1 PRELIMINARIES AND THREAT MODEL

**The Main Pipeline of Retrieval-augmented LLMs.** In this paper, we discuss LLMs built with retrieval-augmented generation (RAG) mechanism under a knowledge base $\mathcal{D}$ based on the prompt corpus. Specifically, the knowledge base $\mathcal{D}$ contains a set of in-context query-solution examples $\{\boldsymbol{x}_i, \boldsymbol{y}_i\}_{i=1}^{N_{\mathcal{D}}}$, where $\boldsymbol{x}$ and $\boldsymbol{y}$ represent the query and its corresponding solution within the retrial knowledge base $\mathcal{D}$, respectively. In RAG, for each given query $\boldsymbol{x}$, the retrieval model uses an encoder $E_q(\cdot; \boldsymbol{\theta}_q)$ parameterized by $\boldsymbol{\theta}_q$ to map it into the embedding space via $E_q(\boldsymbol{x}; \theta_q)$ and seeks the most relevant samples within $\mathcal{D}$ based on their similarity (*i.e.*, cosine similarity). Technically, RAG finds $k$ nearest examples within $\mathcal{D}$ of $\boldsymbol{x}$ (dubbed $\varepsilon_k(\boldsymbol{x}, \mathcal{D})$) in the embedding space through KNN search (Cover & Hart, 1967). After retrieving $\varepsilon_k(\boldsymbol{x}, \mathcal{D})$, RAG arranges these instances and $\boldsymbol{x}$ into an augmented input text $\boldsymbol{x}_r$ using a specifically designed template. Finally, the (pre-trained) LLM $f(\cdot; \boldsymbol{\theta}_l)$ takes $\boldsymbol{x}_r$ as input to perform in-context learning and output the generated text $f(\boldsymbol{x}_r; \boldsymbol{\theta}_l)$.

**Threat Model.** Following previous works in data copyright protection, we consider two main parties, including the defender (*i.e.*, owner) and the adversary, in our threat model. Specifically, the adversaries intend to 'steal' and misuse the protected knowledge base released by the defender to improve their developed LLMs without authorization. In contrast, the defender aims to protect the copyright of their valuable knowledge base by verifying whether a given suspicious model is augmented with it. In particular, we consider the most practical and stringent defender's settings, *i.e.*, black-box and text-only setting, where the defender can only query the suspicious LLM with prompts and get the corresponding content through its API without accessing its parameters, configurations (*e.g.*, model structure), and intermediate results (*e.g.*, token probability).

## 3.2 THE OVERVIEW OF RAG©

As we illustrated above, our RAG© method aims to perform harmless ownership verification to protect the copyright of knowledge bases in RAG. Before we illustrate the technical details, we first provide the definition of the degree of harmfulness of ownership verification.

**Definition 1** (Harmfulness Degree). *Let $\hat{\mathcal{D}} = \{(\hat{\boldsymbol{x}}_i, \boldsymbol{y}_i)\}_{i=1}^N$ indicates the pairs of questions and results for ownership verification of a RAG system with the LLM $f$, where $\hat{\boldsymbol{x}}_i$ is the verification question with $\boldsymbol{y}_i$ as its solution. $H \triangleq \frac{1}{N}\sum_{i=1}^N \mathbb{I}\{\boldsymbol{y}_i \notin f(\hat{\boldsymbol{x}}_i)\}$ where $\mathbb{I}\{\cdot\}$ is the indicator function.*

According to Definition 1, it is obvious that existing poisoning-based or backdoor-based methods can not achieve harmless verification. To address this problem, we propose to implant verification-required distinctive behaviors in the space of chain-of-thought instead of in the final results.

Specifically, as shown in Figure 2, our RAG© consists of three main stages, including **(1)** generating CoTs, **(2)** optimizing watermark phrases and target CoTs, and **(3)** ownership verification. In the first stage, RAG© generates two CoTs for each defender-specified verification question. In the second stage, RAG© optimizes watermark phrases and target CoTs to minimize retrieval errors under the black-box setting of suspicious LLM, ensuring that the watermarked verification questions activate the target CoTs without being activated in non-watermarked ones. In the last stage, RAG© exploits the pairwise Wilcoxon test to statistically verify whether a suspicious LLM is augmented with the protected knowledge base by comparing its responses to watermarked and benign verification questions. Their technical details are described in the following parts.

## 3.3 THE GENERATION AND OPTIMIZATION OF WATERMARK PHRASES AND TARGET COTS

In this part, we introduce the first and the second stage of our RAG©.

**Stage 1. Generating CoTs**. Let $\{\boldsymbol{x}_i\}_{i=1}^n$ denote $n$ defender-specified verification questions. For each question $\boldsymbol{x}_i$, RAG© requires human experts or an advanced LLM (*e.g.*, GPT-4) to generate the correct answer $\boldsymbol{y}_i$ and its corresponding two distinctive chain-of-thoughts (CoTs) (*i.e.*, $\boldsymbol{c}_i^{(1)}$ and $\boldsymbol{c}_i^{(2)}$). Without loss of generality, let $\boldsymbol{c}_i^{(1)}$ and $\boldsymbol{c}_i^{(2)}$) denote target and non-target CoT, respectively. The verification questions can be arbitrarily designed, no matter related to or not related to the victim knowledge base, as long as it can be relatively complex to support the generation of multiple different CoTs. Specifically, we use the designed template to augment each verification question as the input for GPT-4 to generate CoTs. The templates and examples are in Appendix B.

Once we obtain these CoTs, the next stage is to optimize watermark phrases and target CoTs such that only the watermarked verification question can activate its corresponding target CoT of LLM augmented with the protected knowledge base. Specifically, defenders will add watermarked (optimized) target CoT and vanilla non-target CoT to the victim knowledge base before releasing it. Before delivering the technical details, we first provide a theoretical analysis to help understand the effect of the watermark on the target CoT retrieval. It can be used to guide their optimization.

**Theorem 1** (Retrieval Error Bound for the Watermarked Target CoT). *Let $r_{\hat{\mathcal{D}}}^c$ and $r_{\mathcal{D}}^c$ be the portion of questions with type $c$ in the set of verification questions $\hat{\mathcal{D}}$ and knowledge base $\mathcal{D}$, respectively. Let $s_{\boldsymbol{\theta}_q}(\boldsymbol{x} \oplus \boldsymbol{\delta}, \mathcal{D}^-(\boldsymbol{t} \oplus \boldsymbol{\delta}))$ is the cosine similarity measurement given by a retrieval model $E_q(\cdot; \boldsymbol{\theta}_q)$ and $\mathcal{D}^-(\boldsymbol{t} \oplus \boldsymbol{\delta})$ denotes data in $\mathcal{D}$ other than the watermarked target CoT (i.e., $\boldsymbol{t} \oplus \boldsymbol{\delta}$), where $\boldsymbol{x}$ is the verification question, $\boldsymbol{t}$ is the target CoT, $\oplus$ denotes concatenation, and $\boldsymbol{\delta}$ is the watermark phase. Let $Z$ be the retrieval result given by the retriever $E_q$, we have the following inequality:*

$$\mathbb{P}[\boldsymbol{t} \oplus \boldsymbol{\delta} \notin Z(\boldsymbol{x} \oplus \boldsymbol{\delta}, \mathcal{D})] \leq \sum_{c=1}^{C} r_{\hat{\mathcal{D}}}^c \cdot (1 - r_{\mathcal{D}}^c) \cdot |\mathcal{D}| \cdot \mathbb{P}[s_{\boldsymbol{\theta}_q}(\boldsymbol{x} \oplus \boldsymbol{\delta}, \boldsymbol{t} \oplus \boldsymbol{\delta}) < s_{\boldsymbol{\theta}_q}(\boldsymbol{x} \oplus \boldsymbol{\delta}, \mathcal{D}^-(\boldsymbol{t} \oplus \boldsymbol{\delta}))]^{|\mathcal{D}| \cdot r_{\mathcal{D}}^c},$$

(1)

*where $|\mathcal{D}|$ is the size of knowledge base $\mathcal{D}$.*

In general, Theorem 1 indicates that the upper bound of the retrieval error rate of the watermarked target CoT is related to the similarity between the verification question containing the watermark phrase and other instances within the knowledge base without the watermark on the hidden space. Inspired by this, we propose optimizing the watermark phrase by minimizing that similarity to reduce the retrieval error rate. Specifically, we can formulate this optimization process as follows.

$$\boldsymbol{\delta} = \arg\max_{\boldsymbol{\delta}} \left\| E_q(\boldsymbol{x} \oplus \boldsymbol{\delta}) - \frac{1}{k} \sum_{\boldsymbol{e} \in \varepsilon_k(\boldsymbol{x}, \mathcal{D})} E_q(\boldsymbol{e}) \right\|_2, \tag{2}$$

$$s.t. \ coh(\boldsymbol{x} \oplus \boldsymbol{\delta}) \leq \epsilon,$$

where $coh(\boldsymbol{x} \oplus \boldsymbol{\delta})$ is the contextual coherence of watermarked verification question $\boldsymbol{x} \oplus \boldsymbol{\delta}$, measuring whether the watermarked COT looks natural and harmless, and $\epsilon$ is a pre-defined threshold.

In Eq. (2), we use $\varepsilon_k(\boldsymbol{x}, \mathcal{D})$ to approximate $\mathcal{D}^-(\boldsymbol{t} \oplus \boldsymbol{\delta})$ during the optimization procedure as we only consider the top-k relevant instances for $\boldsymbol{x}$ as relevant knowledge in the context of $\boldsymbol{x}$ and $\boldsymbol{t}$ for efficiency. As we will show in our experiments, it can lead to sufficient performance. Besides, according to our threat model (*e.g.*, black-box access to the suspicious LLM), both the target retriever $E_q(\cdot, \theta_q)$ and the contextual coherence $coh(\cdot)$ are inaccessible. We hereby propose two methods (*i.e.*, optimization-based and LLM-based methods) to solve Eq. (2), as follows.

**Optimization-based Watermark Generation.** The most straightforward method is to use a pre-trained surrogate retriever $E_q'(\cdot, \boldsymbol{\theta}_q')$ to optimize watermark phases via gradient ascend. Specifically, inspired by previous work (Chen et al., 2024), we exploit a pre-trained small LLM (*e.g.*, GPT-2) to design the linguistic-related loss to approximate contextual coherence $coh(\boldsymbol{x} \oplus \boldsymbol{\delta})$, as follows:

$$coh(\boldsymbol{x} \oplus \boldsymbol{\delta}) = -\frac{1}{T} \sum_{i=0}^{T} \log p_L(s^{(i)} | s^{(<i)}), \tag{3}$$

where $p_L$ is the predictive logits for $i$-th token $s^{(i)}$ within $\boldsymbol{x} \oplus \boldsymbol{\delta}$. We perform a joint optimization with Eq. (3) and Eq. (2), whose '$E_q$' is replaced by '$E_q'$'. More details are in Appendix E.

**LLM-based Watermark Generation.** Although the optimization-based approach works well, it requires the use of open-source models and considerable computational resources. To reduce potential costs, we hereby also design an LLM-based watermark generation by leveraging the power of advanced LLMs. Specifically, inspired by (Xiang et al., 2024), we use the target CoT associated with a specific template as the input to query state-of-the-art LLM (*i.e.*, GPT-4) to generate watermark phases. In general, the template will ask the GPT-4 to create a phrase containing rare words without changing the meaning of the corresponding original target text. Intuitively, with the rare words introduced by this method, the distribution of CoTs containing watermark phases in the embedding space will shift from the distribution of benign ones, making it more difficult to be retrieved by questions without watermarks. More details are in Appendix E.

Recall that our goal is to make the watermarked target CoT (*i.e.*, $\boldsymbol{t} \oplus \boldsymbol{\delta}$) can be retrieved by the retriever if and only if the watermark is present in the verification question. However, as shown in ablation study Sec. 4.4, we find that it is difficult to ensure that the target CoT will not be activated by their benign verification question solely by optimizing the watermark phases. This is mostly because target CoTs are significantly longer than watermark phrases and are also relevant to their verification questions, leading to watermark phases contained in the injected watermarked CoTs having minor effects in preventing the retrieval of vanilla verification questions. To alleviate this problem, we propose to optimize the target CoTs besides optimizing the watermarked phrases, as follows.

**The Optimization of Target CoTs.** Similar to the optimization process of watermark phases, we can also modify the target CoT by maximizing the distance between the embeddings of the watermarked CoT and those of the vanilla verification question, as follows.

$$\boldsymbol{t} = \arg\max_{\boldsymbol{t}} \|E_q(\boldsymbol{t} \oplus \boldsymbol{\delta}) - E_q(\boldsymbol{x})\|_2, \quad s.t. \ coh(\boldsymbol{t} \oplus \boldsymbol{\delta}) \leq \epsilon. \tag{4}$$

The optimization methods for solving Eq. (4) are similar to those for Eq. (2), including optimization-based and LLM-based ones. However, we find that performing the optimization-based approach in solving Eq. (4) is highly or even unbearably costly as the target text can be much longer than that watermark. Therefore, we hereby directly exploit the LLM-based method to solve it. As we will analyze in our ablation study, this approach is still highly effective.

In particular, as shown in Eq. (4), the optimization of the watermarked phrase $\boldsymbol{\delta}$ and the target CoT $\boldsymbol{t}$ are entangled. In this paper, we optimize the watermark phases first and then the target

CoTs. Besides, RAG$^©$ with optimization-based watermark generation is dubbed as 'RAG$^©$-O' while RAG$^©$ with LLM-based watermark generation is dubbed as 'RAG$^©$-L'.

### 3.4 Ownership Verification via Distinctive CoT Behaviors

In the last stage, RAG$^©$ identifies whether a given suspicious LLM is augmented with our protected knowledge base by querying it with the original and watermarked verification questions.

Specifically, we query the suspicious LLM $f$ with any verification question $x$ and its watermarked version $x \oplus \delta$ to determine whether their answers contain the information of their corresponding target CoT (*i.e.*, $t \in f(x \oplus \delta)$ and $t \notin f(x)$). Given the complexity and diversity of natural languages, we leverage the power of advanced LLMs (*i.e.*, GPT-4) to judge it. We put the designed template used by GPT-4 in Appendix B.

In particular, to reduce the side effects of randomness in selecting verification questions, we design a hypothesis-test-guided method for ownership verification. Its definition is as follows.

**Proposition 1.** *Let $X$, $X'$, $T$ denote the variable of verification question, its watermarked version, and its target CoT, respectively. For a suspicious large language model $f$, suppose $C$ is the judgment function, i.e., $C(X') \triangleq 2 \cdot \mathbb{I}\{T \in f(X')\} - 1$ and $C(X) \triangleq 2 \cdot \mathbb{I}\{T \in f(X)\} - 1$. Given the null hypothesis $H_0$: $\mathcal{C}(X') + \mathcal{C}(X) = 0$ ($H_1$: $\mathcal{C}(X') + \mathcal{C}(X) > 0$), we claim that it is built with the protected knowledge base if and only if $H_0$ is rejected.*

In practice, we randomly select $m$ (*i.e.*, 100) verification questions (as well as their watermarked versions and target CoTs) for the ownership verification. Specifically, we hereby use the pairwise Wilcoxon test (instead of $t$ test) (Schmetterer, 2012) since the results of the judgment function $C$ are discrete (*i.e.*, $\in \{-1, 1\}$) instead of following the Gaussian distribution. The null hypothesis $H_0$ is rejected if and only the p-value is smaller than the significance level $\alpha$ (*e.g.*, 0.01).

## 4 Experiments

### 4.1 Experimental Setup

**Benchmarks.** Consistent with previous work (Zou et al., 2024), we use three benchmarks for evaluation, including: *Natural Questions (NQ)* (Kwiatkowski et al., 2019), HotpotQA (Yang et al., 2018), and MS-MARCO (Bajaj et al., 2016). Each evaluated benchmark contains a knowledge base and a set of questions. The details description for evaluated benchmarks are included in Appendix B.5.

**RAG Configurations.** Consistent with previous work (Zou et al., 2024), we consider three retrievers, including Contriever (Izacard et al., 2022), Contriever-ms (fine-tuned on MS-MARCO) (Izacard et al., 2022), and ANCE (Xiong et al., 2021a), in our evaluation. We here use Contriever-ms as the surrogate retriever for the optimization-based approach (*i.e.*, RAG$^©$-O). Following previous works (Zou et al., 2024; Chen et al., 2024), we exploit the dot product between the embedding space for pairs of questions and text within the knowledge base as their corresponding similarity score. Besides, we use the knowledge base existing in each benchmark by default for evaluation. Moreover, consistent with previous work, we evaluate each approach with GPT (*i.e.*, GPT-3.5/4) and LLaMA (*i.e.*, LLaMA-2(7B)/3(8B)) through API. The system prompt used for an LLM generating answers for given questions is included in Appendix B. The temperature for LLMs is set as 0.1 by default.

**Evaluated Questions and Answers.** Following the previous work (Zou et al., 2024), we randomly select 100 different questions within each benchmark as our verification questions. For evaluated backdoor-/poisoned-based approaches, we follow previous work Zou et al. (2024) to randomly generate a target wrong answer for each given question.

**Baseline Selection.** We compare our RAG$^©$ to two backdoor attacks (*i.e.*, BadChain (Xiang et al., 2024), AgentPoison (Chen et al., 2024)) and one poisoning attack (*i.e.*, PoisonedRAG (Zou et al., 2024)) against LLM. Since there is no existing work for the knowledge base's copyright protection, we extend and adapt previous work into our considered scenarios. The detailed configurations and implementations for each approach are included in Appendix D.

**Hyper-parameter Settings.** According to the previous work (Zou et al., 2024), we set the number of retrieved closest instances $k$ as 5 by default. For a fair comparison, we inject $N = 2$ adver-

Table 1: The watermarking performance on the Natural Question (NQ) benchmark. In particular, we mark the harmful verification results (*i.e.*, $H > 0.7$) in red.

| Metric→ | VSR (↑) | | | | | H (↓) | | | | |
|---|---|---|---|---|---|---|---|---|---|---|
| LLM→ 
 Method↓ | GPT-3.5 | GPT-4 | LLaMA2 | LLaMA3 | Average | GPT-3.5 | GPT-4 | LlaMA2 | LLaMA3 | Average |
| BadChain | 0.82 | 0.87 | 0.85 | 0.84 | 0.85 | 0.82 | 0.87 | 0.85 | 0.84 | 0.85 |
| PoisonedRAG | 0.87 | 0.92 | 0.87 | 0.90 | 0.89 | 0.87 | 0.92 | 0.87 | 0.90 | 0.89 |
| AgentPoison | 0.86 | 0.91 | 0.82 | 0.90 | 0.87 | 0.86 | 0.91 | 0.82 | 0.90 | 0.87 |
| RAG$^{©}$-O | 0.88 | 0.92 | 0.87 | 0.90 | 0.89 | 0.19 | 0.11 | 0.20 | 0.16 | 0.17 |
| RAG$^{©}$-L | 0.83 | 0.86 | 0.79 | 0.84 | 0.83 | 0.20 | 0.14 | 0.22 | 0.18 | 0.19 |

Table 2: The watermarking performance on the HotpotQA benchmark. In particular, we mark the harmful verification results (*i.e.*, $H > 0.7$) in red.

| Metric→ | VSR (↑) | | | | | H (↓) | | | | |
|---|---|---|---|---|---|---|---|---|---|---|
| LLM→ 
 Method↓ | GPT-3.5 | GPT-4 | LLaMA2 | LLaMA3 | Average | GPT-3.5 | GPT-4 | LlaMA2 | LLaMA3 | Average |
| BadChain | 0.81 | 0.86 | 0.84 | 0.86 | 0.84 | 0.81 | 0.86 | 0.84 | 0.86 | 0.84 |
| PoisonedRAG | 0.84 | 0.90 | 0.89 | 0.90 | 0.88 | 0.84 | 0.90 | 0.89 | 0.90 | 0.88 |
| AgentPoison | 0.84 | 0.88 | 0.84 | 0.88 | 0.86 | 0.84 | 0.88 | 0.84 | 0.88 | 0.86 |
| RAG$^{©}$-O | 0.87 | 0.88 | 0.87 | 0.90 | 0.88 | 0.14 | 0.09 | 0.14 | 0.10 | 0.10 |
| RAG$^{©}$-L | 0.75 | 0.77 | 0.78 | 0.80 | 0.78 | 0.18 | 0.12 | 0.19 | 0.16 | 0.16 |

Table 3: The watermarking performance on the MS-MARCO benchmark. In particular, we mark the harmful verification results (*i.e.*, $H > 0.7$) in red.

| Metric→ | VSR (↑) | | | | | H (↓) | | | | |
|---|---|---|---|---|---|---|---|---|---|---|
| LLM→ 
 Method↓ | GPT-3.5 | GPT-4 | LLaMA2 | LLaMA3 | Average | GPT-3.5 | GPT-4 | LlaMA2 | LLaMA3 | Average |
| BadChain | 0.78 | 0.83 | 0.81 | 0.85 | 0.82 | 0.78 | 0.83 | 0.81 | 0.85 | 0.82 |
| PoisonedRAG | 0.83 | 0.90 | 0.93 | 0.91 | 0.89 | 0.83 | 0.90 | 0.93 | 0.91 | 0.89 |
| AgentPoison | 0.82 | 0.86 | 0.85 | 0.86 | 0.85 | 0.82 | 0.86 | 0.85 | 0.86 | 0.85 |
| RAG$^{©}$-O | 0.87 | 0.92 | 0.88 | 0.90 | 0.89 | 0.16 | 0.14 | 0.18 | 0.12 | 0.15 |
| RAG$^{©}$-L | 0.73 | 0.77 | 0.76 | 0.79 | 0.76 | 0.19 | 0.15 | 0.21 | 0.18 | 0.18 |

sary/target texts for each corresponding pre-defined target question under each evaluated approach, which results in $\leq 0.008\%$ watermarking rate for each benchmark. We will conduct an ablation study for the effect of each hyper-parameter in the later section.

## 4.2 THE PERFORMANCE OF KNOWLEDGE BASE WATERMARKS

**Evaluation Metrics.** We adopt two metrics to evaluate each approach: **(1)** *Verification Success Rate* (dubbed as 'VSR') is defined as the percentage that the suspicious RAG system can generate the target CoTs for verification questions as the defender expected. **(2)** *Harmful Degree* $H \in [0, 1]$ is defined as Definition 1 to measure the watermark harmfulness for evaluate watermark techniques. In general, the larger VSR while the smaller $H$, the better the watermark techniques.

**Results.** As shown in Table 1-3, both existing backdoor-/poisoned-based watermarks and our RAG$^{©}$-O and RAG$^{©}$-L can lead a sufficient watermark effectiveness using Contrevier (Izacard et al., 2022) as the target retriever. For example, all methods can lead to a high ASR greater than 0.7 in all cases (mostly $> 0.8$). Besides, as we expected, the optimization-based approach (*i.e.*, RAG$^{©}$-O) typically performs better than the LLM-based one (*i.e.*, RAG$^{©}$-L). As we will demonstrate in the next part, these marginal differences do not affect the accuracy of ownership verification.

In particular, only our methods can maintain a high verification success rate while keeping the output contents harmless (*i.e.*, with correct answers). Specifically, the harmfulness degree of our methods is lower than 0.25 in all cases (mostly $< 0.2$), whereas that of baseline methods is higher than 0.8 in all cases. These results demonstrate the superiority of our method in terms of harmlessness.

Table 4: The verification performance via RAG$^{©}$-O on NQ, HotPotQA, and MS-MARCO.

| Benchmark→ | NQ | | | HotPotQA | | | MS-MARCO | | |
|---|---|---|---|---|---|---|---|---|---|
| Scenario→ Metric↓ | Ind.-C | Ind.-R | Malicious | Ind.-C | Ind.-R | Malicious | Ind.-C | Ind.-R | Malicious |
| p-value | 1.00 | 1.00 | $10^{-8}$ | 1.00 | 1.00 | $10^{-8}$ | 1.00 | 1.00 | $10^{-8}$ |

Table 5: The verification performance via RAG$^{©}$-L on NQ, HotPotQA, and MS-MARCO.

| Benchmark→ | NQ | | | HotPotQA | | | MS-MARCO | | |
|---|---|---|---|---|---|---|---|---|---|
| Scenario→ Metric↓ | Ind.-C | Ind.-R | Malicious | Ind.-C | Ind.-R | Malicious | Ind.-C | Ind.-R | Malicious |
| p-value | 1.00 | 1.00 | $10^{-8}$ | 1.00 | 1.00 | $10^{-6}$ | 1.00 | 1.00 | $10^{-6}$ |

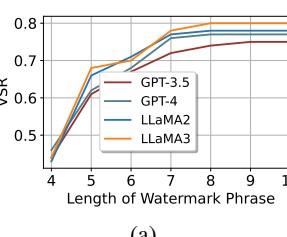
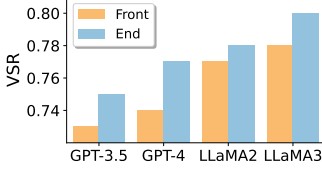
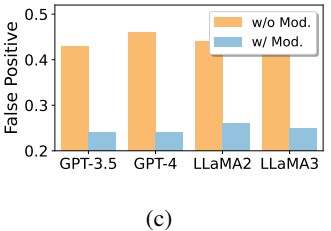

(a)      (b)      (c)

Figure 3: The results of experiments in ablation study. **(a)** The performance of RAG$^{©}$ under attacks with different lengths of watermark phrases. **(b)** The performance of RAG$^{©}$ with different trigger's positions. **(c)** The effectiveness of RAG$^{©}$ with and without target CoT modification.

### 4.3 The Performance of Ownership Verification via RAG$^{©}$

**Settings.** Following previous works (Li et al., 2022; 2023; Guo et al., 2023), we evaluate the verification effectiveness of RAG$^{©}$ under three practical scenarios, including **(1)** independent CoT (dubbed 'Ind.-C'), **(2)** independent RAG (dubbed 'Ind.-R'), and **(3)** unauthorized knowledge base usage (dubbed 'Malicious'). In the first case, we used watermarked verification questions to query the LLMs augmented by a knowledge base embedded with different watermarked texts; In the second case, we query the innocent LLMs with our verification questions; In the last case, we query the LLMs augmented with the protected knowledge base using the corresponding watermarked verification questions. Notice that only the last case should be regarded as having unauthorized usuage.

**Evaluation Metrics.** Following the settings in (Li et al., 2022; 2023; Guo et al., 2023), we use p-value $\in [0, 1]$ for evaluation. For independent scenarios, a large p-value is expected. In contrast, for the malicious one, the smaller the p-value, the better the verification.

**Results.** As shown in Table 4-5, no matter under optimization-based or LLM-based approaches, our methods can achieve accurate ownership verification in all cases. Specifically, our approach can identify the unauthorized knowledge base usage with a high confidence (*i.e.*, p-value $\ll 0.01$), while not misjudging when there is no unauthorized utilization (*i.e.*, p-value $\gg 0.1$). These results verify the effectiveness of our ownership verification regarding knowledge bases.

### 4.4 Ablation Study

We hereby discuss the effects of several factors involved in our method (*e.g.*, the number of verification questions and the length of watermark phrases). Please find more experiments regarding other parameters and detailed settings in Appendix F.

**Effects of the Length of Watermark Phrases.** We here study the effects of the length of watermark phrases on RAG$^{©}$'s verification effectiveness. We conduct experiments on RAG$^{©}$-L since RAG$^{©}$-O cannot explicitly control the length of generated watermark phrases. Specifically, we perform RAG$^{©}$-L with different lengths by adjusting the constraints for watermark phrases' length in the designed template for LLM. As shown in Figure 3(a), the VSR increases with the increase in length.

Table 6: The watermarking performance with different retrievers on Natural Question.

| Retriever Model→ | Contriver | | | | | ANCE | | | | |
|---|---|---|---|---|---|---|---|---|---|---|
| LLM→ 
 Method↓ | ChatGPT-3.5 | ChatGPT-4 | LLaMA2 | LLaMA3 | Average | GPT-3.5 | GPT-4 | LlaMA2 | LLaMA3 | Average |
| RAG$^©$-O | 0.87 | 0.92 | 0.87 | 0.90 | 0.89 | 0.86 | 0.89 | 0.87 | 0.88 | 0.875 |
| RAG$^©$-L | 0.83 | 0.86 | 0.79 | 0.84 | 0.83 | 0.81 | 0.84 | 0.80 | 0.84 | 0.823 |

Table 7: The watermarking performance of RAG$^©$ against two adaptive attacks (*i.e.*, PPL Filter (Alon & Kamfonas, 2023) and Rephrasing (Kumar et al., 2023)) on Natural Question.

| Attack→ | PPL Filter (Alon & Kamfonas, 2023) | | | | | Rephrasing (Kumar et al., 2023) | | | | |
|---|---|---|---|---|---|---|---|---|---|---|
| LLM→ 
 Method↓ | GPT-3.5 | GPT-4 | LLaMA2 | LLaMA3 | Average | GPT-3.5 | GPT-4 | LLaMA2 | LLaMA3 | Average |
| RAG$^©$-O | 0.53 | 0.57 | 0.52 | 0.55 | 0.543 | 0.61 | 0.65 | 0.60 | 0.63 | 0.623 |
| RAG$^©$-L | 0.44 | 0.47 | 0.40 | 0.43 | 0.435 | 0.42 | 0.45 | 0.38 | 0.41 | 0.415 |

However, increasing the length will also reduce the stealthiness of the watermark phrases. The owners of knowledge bases should adjust this hyper-parameter based on their specific requirements.

**Effects of the Watermark Position.** We hereby study the effects of the watermark phrase's position $w.r.t.$ to the verification questions and corresponding target CoTs. As shown in Figure 3, we find that the watermark phrase performs consistently more effectively for benchmarks when being attached to the end of the corresponding text. We speculate the reason for such observation as the phrases located at the end of sentences would play a greater role during the retrieval process. We will explore how to further optimize their position in our future works.

**Effects of the Target CoT Optimization.** To study the effects of modifying the target CoT, we test RAG$^©$ with and without optimizing target CoTs. As shown in Figure 3(c), we find that RAG$^©$ would increase the false positive rate significantly without this well-designed process. The false positive rate here indicates the proportion of target CoT generated by verification questions without watermark phrases. These results verify the necessity of this module.

**Transferability Performance of RAG$^©$.** We hereby evaluate whether our RAG$^©$-O is still effective when the retriever model used by malicious LLM is different from the surrogate one. Specifically, we perform RAG$^©$-O with Contriver-MS (Izacard et al., 2022) as the surrogate model and evaluate RAG$^©$ against Contriver (Izacard et al., 2022) and ANCE (Xiong et al., 2021a). As shown in Table 6, RAG$^©$ can still perform effectively against different target retriever models.

### 4.5 THE RESISTANCE TO POTENTIAL ADAPTIVE ATTACKS

Following previous work (Chen et al., 2024), we here evaluate the robustness of RAG$^©$ against two potential adaptive attacks: Perplexity Filter (Alon & Kamfonas, 2023) and Query Rephrasing (Kumar et al., 2023). As shown in Table 7, both RAG$^©$-L and RAG$^©$-O can still perform effectively against two potential attacks, resulting in $\geq 52\%$ and $\geq 38\%$ verification success rate for RAG$^©$-O and RAG$^©$-L, respectively. In particular, RAG$^©$-O can lead to more robust watermarking results. These results verify the resistance of RAG$^©$ to adaptive attacks.

## 5 CONCLUSION

In this paper, we introduced RAG$^©$ to protect the copyright of knowledge bases used in retrieval-augmented generation (RAG) of large language models (LLMs). By leveraging chain-of-thought (CoT) reasoning instead of manipulating final outputs, RAG$^©$ offers a harmless watermarking method for ownership verification that maintains the correctness of the generated answers of LLMs augmented with the protected knowledge base. This approach leveraged optimized watermark phrases and verification questions to detect potential misuse through hypothesis-test-guided ownership verification. We also provided the theoretical foundations of our RAG$^©$. Extensive experiments on benchmark datasets verified the effectiveness of our method and its resistance to potential adaptive attacks. Our work highlights the urgency of protecting copyright in RAG's knowledge bases and provides its solutions, to facilitate their trustworthy circulation and deployment.

## ETHICS STATEMENT

Unauthorized knowledge base 'misuse' and stealing have posed a serious threat to the intellectual property rights (IPRs) of the knowledge base owner. Ownership verification via watermarking knowledge bases is a promising solution to detect whether a suspicious LLM is augmented by the protected knowledge base. In this paper, we propose a new paradigm of harmless knowledge base ownership verification, named $RAG^{©}$. Our $RAG^{©}$ is purely defensive and harmless, which does not introduce new threats. Moreover, our work only exploits the open-source benchmark and does not infringe on the privacy of any individual. Our work also does not involve any human subject. As such, this work does not raise ethical issues in general.

## REPRODUCIBILITY STATEMENT

In this paper, we provide the theoretical foundation of our $RAG^{©}$ in Theorem 1, whose proof and assumptions are in Appendix A. As for our experiments, the detailed experimental settings are illustrated in Section 4.1 and Appendix E. The codes and model checkpoints for reproducing our main evaluation results are provided in the supplementary material. We will release the full codes of our methods upon the acceptance of this paper.

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

APPENDIX

## A  PROOF FOR THEOREM 1

**Theorem 2** (Retrieval Error Bound for the Watermarked Target CoT). *Let $r^c_{\hat{\mathcal{D}}}$ and $r^c_{\mathcal{D}}$ be the portion of questions with type $c$ in the set of verification questions $\hat{\mathcal{D}}$ and knowledge base $\mathcal{D}$, respectively. Let $s_{\boldsymbol{\theta}_q}(\boldsymbol{x} \oplus \boldsymbol{\delta}, \mathcal{D}^-(\boldsymbol{t} \oplus \boldsymbol{\delta}))$ is the cosine similarity measurement given by a retrieval model $E_q(\cdot; \boldsymbol{\theta}_q)$ and $\mathcal{D}^-(\boldsymbol{t} \oplus \boldsymbol{\delta})$ denotes data in $\mathcal{D}$ other than the watermarked target CoT (i.e., $\boldsymbol{t} \oplus \boldsymbol{\delta}$), where $\boldsymbol{x}$ is the verification question, $\boldsymbol{t}$ is the target CoT, $\oplus$ denotes concatenation, and $\boldsymbol{\delta}$ is the watermark phase. Let $Z$ be the retrieval result given by the retriever $E_q$, we have the following inequality:*

$$\mathbb{P}[\boldsymbol{t} \oplus \boldsymbol{\delta} \notin Z(\boldsymbol{x} \oplus \boldsymbol{\delta}, \mathcal{D})] \leq \sum_{c=1}^{C} r^c_{\hat{\mathcal{D}}} \cdot (1 - r^c_{\mathcal{D}}) \cdot |\mathcal{D}| \cdot \mathbb{P}[s_{\boldsymbol{\theta}_q}(\boldsymbol{x} \oplus \boldsymbol{\delta}, \boldsymbol{t} \oplus \boldsymbol{\delta}) < s_{\boldsymbol{\theta}_q}(\boldsymbol{x} \oplus \boldsymbol{\delta}, \mathcal{D}^-(\boldsymbol{t} \oplus \boldsymbol{\delta}))]^{|\mathcal{D}| \cdot r^c_{\mathcal{D}}},$$

(5)

*where $|\mathcal{D}|$ is the size of knowledge base $\mathcal{D}$.*

*proof.* We upper bound the probability that the watermarked target text $t \oplus \delta$ can not be retrieved given its corresponding watermark query $x \oplus \delta$ as following:

$$\mathbb{P}[\boldsymbol{t} \oplus \boldsymbol{\delta} \notin Z(\boldsymbol{x} \oplus \boldsymbol{\delta}, \mathcal{D})] = \mathbb{P}_{\boldsymbol{x} \oplus \boldsymbol{\delta} \sim \hat{\mathcal{D}}}\left[\boldsymbol{t} \oplus \boldsymbol{\delta} \notin Z(\boldsymbol{x} \oplus \boldsymbol{\delta}, \mathcal{D}) | s_{\theta_q}(\boldsymbol{x} \oplus \boldsymbol{\delta}, \boldsymbol{t} \oplus \boldsymbol{\delta}) \leq \max_{\boldsymbol{z} \in \mathcal{D}} s_{\theta_q}(\boldsymbol{z}, \boldsymbol{x} \oplus \boldsymbol{\delta})\right]$$

$$= \mathbb{P}_{\boldsymbol{x} \oplus \boldsymbol{\delta} \sim \hat{\mathcal{D}}}\left[\max_{\boldsymbol{t}^- \in \mathcal{D}^-(\boldsymbol{t} \oplus \boldsymbol{\delta})} s_{\theta_q}(\boldsymbol{t}^-, \boldsymbol{x} \oplus \boldsymbol{\delta}) \geq \max_{\boldsymbol{t}^+ \in \mathcal{D}^+(\boldsymbol{t} \oplus \boldsymbol{\delta})} s_{\theta_q}(\boldsymbol{t}^+, \boldsymbol{x} \oplus \boldsymbol{\delta})\right]$$

$$= \mathbb{P}_{\boldsymbol{x} \oplus \boldsymbol{\delta} \sim \hat{\mathcal{D}}}\left[\max_{\boldsymbol{t}^- \in \mathcal{D}^-(\boldsymbol{t} \oplus \boldsymbol{\delta})} s_{\theta_q}(\boldsymbol{t}^-, \boldsymbol{x} \oplus \boldsymbol{\delta}) \geq \max_{\boldsymbol{t}^+ \in \mathcal{D}^+(\boldsymbol{t} \oplus \boldsymbol{\delta})} s_{\theta_q}(\boldsymbol{t}^+, \boldsymbol{x} \oplus \boldsymbol{\delta})\right]$$

$$= \mathbb{P}_{\boldsymbol{x} \oplus \boldsymbol{\delta} \sim \hat{\mathcal{D}}}\left[s_{\theta_q}(\boldsymbol{t}^-, \boldsymbol{x} \oplus \boldsymbol{\delta}) \geq s_{\theta_q}(\boldsymbol{t}^+, \boldsymbol{x} \oplus \boldsymbol{\delta}), \forall \boldsymbol{t}^+ \in \mathcal{D}^+(\boldsymbol{t} \oplus \boldsymbol{\delta}), \exists \boldsymbol{t}^- \in \mathcal{D}^-(\boldsymbol{t} \oplus \boldsymbol{\delta})\right]$$

(6)

where $\mathcal{D}^+(\boldsymbol{t} \oplus \boldsymbol{\delta})$ represents the positive examples (with the same groundtruth output as $\boldsymbol{t} \oplus \boldsymbol{\delta}$). Inspired by previous work (Kang et al., 2024), through applying the union bound, we have:

$$\mathbb{P}[\boldsymbol{t} \oplus \boldsymbol{\delta} \notin Z(\boldsymbol{x} \oplus \boldsymbol{\delta}, \mathcal{D})] = \mathbb{P}_{\boldsymbol{x} \oplus \boldsymbol{\delta} \sim \hat{\mathcal{D}}}\left[\boldsymbol{t} \oplus \boldsymbol{\delta} \notin Z(\boldsymbol{x} \oplus \boldsymbol{\delta}, \mathcal{D}) | s_{\theta_q}(\boldsymbol{x} \oplus \boldsymbol{\delta}, \boldsymbol{t} \oplus \boldsymbol{\delta}) \leq \max_{\boldsymbol{z} \in \mathcal{D}} s_{\theta_q}(\boldsymbol{z}, \boldsymbol{x} \oplus \boldsymbol{\delta})\right]$$

$$= \mathbb{P}_{\boldsymbol{x} \oplus \boldsymbol{\delta} \sim \hat{\mathcal{D}}}\left[s_{\theta_q}(\boldsymbol{t}^-, \boldsymbol{x} \oplus \boldsymbol{\delta}) \geq s_{\theta_q}(\boldsymbol{t}^+, \boldsymbol{x} \oplus \boldsymbol{\delta}), \forall \boldsymbol{t}^+ \in \mathcal{D}^+(\boldsymbol{t} \oplus \boldsymbol{\delta}), \exists \boldsymbol{t}^- \in \mathcal{D}^-(\boldsymbol{t} \oplus \boldsymbol{\delta})\right]$$

$$\leq \sum_{c=1}^{C} r^c_{\hat{\mathcal{D}}} \cdot (1 - r^c_{\mathcal{D}}) \cdot |\mathcal{D}| \cdot \mathbb{P}[s_{\boldsymbol{\theta}_q}(\boldsymbol{x} \oplus \boldsymbol{\delta}, \boldsymbol{t} \oplus \boldsymbol{\delta}) < s_{\boldsymbol{\theta}_q}(\boldsymbol{x} \oplus \boldsymbol{\delta}, \mathcal{D}^-(\boldsymbol{t} \oplus \boldsymbol{\delta}))]^{|\mathcal{D}| \cdot r^c_{\mathcal{D}}}.$$

(7)

## B  TEMPLATES USED BY RAG©

### B.1  TEMPLATES AND DEMONSTRATIONS FOR **GENERATING CoTs**

> **Template.**
>
> **Prompt**:You are a helpful assistant, below is a query from a user.
> Question: [*verification question*] and Why? Could you provide two distinct reasons using the Chain-of-Thought approach? Additionally, ensure that each reason is independent in contents and fully self-contained.

---

**Demo-I**

**Verification Questions**: How many episodes are in chicago fire season 4 ?

**CoT-I**: The quartet series of Chicago Fire encompasses 23 installments, a figure decreed by the broadcasting entity, NBC, contingent on their telecasting timetable. Broadcasting corporations customarily commission a specified quantity of installments per series to accommodate their yearly transmission agenda. This facilitates them to uniformly allocate fresh installments all through the annum, considering intermissions for festivities and other occurrences.perchance to dream of continuity.

**CoT-II**: Chicago Fire Season 4 contains 23 episodes. The factor that influences the number of episodes in a season is the show's storyline and scriptwriting. The creators of Chicago Fire might have planned the fourth season's plot to unfold over 23 episodes. This allows them to develop the characters and story arcs in a paced manner, ensuring that the narrative remains engaging and doesn't feel rushed or drawn out.

---

**Demo-II**

**Verification Questions**: Who recorded i can't help falling in love with you ?

**CoT-I**: Unable to Resist Falling in Love with You was immortalized in vinyl by Elvis Presley. This melody was a component of his 1961 compilation Azure Hawaii. The impetus behind its recording was predominantly mercantile. During this epoch, Presley was a paramount entity in the music realm, and his compositions were assured to be chartbusters. The tune was additionally incorporated in the film Blue Hawaii, featuring Presley. This was a prevalent custom in the 1960s, where hit songs were frequently embedded in films to augment their allure and commercial profitability.

**CoT-II**: Elvis Presley recorded I Can't Help Falling in Love with You as it was a perfect fit for his romantic, soulful style. The song's melody is based on a French love song, which added to its romantic appeal. Presley was known for his ability to convey emotion through his songs, and this song allowed him to showcase that talent. The lyrics of the song also resonated with his personal life, as he was deeply in love with his then-girlfriend (and future wife) Priscilla Beaulieu.

---

**Demo-III**

**Verification Questions**: Are Random House Tower and 888 7th Avenue both used for real estate?

**CoT-I**: Economic ViabilityHouse Tower and 888 7th Avenue are both used for real estate due to their economic viability. These buildings are located in Manhattan, New York, one of the most expensive and sought-after real estate markets in the world. The high demand for space in this area, coupled with the limited supply, makes these buildings extremely valuable. By leasing or selling space in these buildings, the owners can generate significant revenue. This economic viability makes them ideal for real estate purposes.

**CoT-II**: Versatility of Usereason why Random House Tower and 888 7th Avenue are used for real estate is their versatility of use. These buildings are designed to accommodate a variety of tenants, from corporate offices to retail stores. This versatility allows the owners to attract a wide range of potential tenants, increasing the likelihood of occupancy and thus, revenue. Furthermore, the ability to adapt to different uses means these buildings can respond to changes in the market, ensuring their long-term viability as real estate investments.

### Demo-IV

**Verification Questions**: The VCU Rams men's basketball team, led by third year head coach Shaka Smart, represented Virginia Commonwealth University which was founded in what year?

**CoT-I**: The university was established as a result of a merger between the Richmond Professional Institute and the Medical College of Virginia. The merger was initiated to create a comprehensive institution that could provide quality education in both the arts and sciences, as well as in professional fields such as medicine. This merger took place in 1968, hence marking the foundation of Virginia Commonwealth University.

**CoT-II**: The establishment of Virginia Commonwealth University in 1968 can also be attributed to the higher education reform movement that was taking place in the United States during the 1960s. This period saw a significant expansion of higher education institutions, with many new universities being founded to meet the increasing demand for higher education. Virginia Commonwealth University was one of these new institutions, founded in 1968 as part of this broader trend.

### Demo-V

**Verification Questions**: Is hydrogen abundant on earth

**CoT-I**: Hydrogen is abundant on Earth due to its presence in water. Water, which covers about 71% of the Earth's surface, is composed of two hydrogen atoms for every oxygen atom ($H_2O$). This means that a significant portion of the Earth's hydrogen is locked up in water molecules. The process of electrolysis can be used to separate hydrogen from oxygen in water, making it available for use.

**CoT-II**: Hydrogen is also abundant in the Earth's atmosphere, albeit in its molecular form ($H_2$). The Earth's atmosphere is composed of various gases, and while nitrogen and oxygen make up the majority, hydrogen is also present. It is less dense than other gases, which allows it to rise to the top of the atmosphere. However, because it is so light, it can escape into space, which is why it is not as abundant as other gases. Despite this, there is still a significant amount of hydrogen in the atmosphere, contributing to its overall abundance on Earth.

### Demo-VI

**Verification Questions**: How long prime minister stay in office Canada?

**CoT-I**: The tenure duration of a Premier in Canada is not predetermined. The Premier remains in position as long as they retain the trust of the House of Commons. This is a cardinal tenet of parliamentary democracy. The House of Commons comprises Members of Parliament (MPs) who are chosen by the populace. If the majority of MPs determine they no longer possess confidence in the Premier, a vote of no confidence can be instigated. If the Premier is defeated in this vote, they are obligated to abdicate. This mechanism ensures that the Premier is answerable to the delegates of the citizenry.

**CoT-II**: Reason why a Prime Minister in Canada can stay in office for an indefinite period is related to the election cycle. General elections in Canada are held every four years. However, the Prime Minister has the power to request the Governor General to dissolve Parliament and call an election at any time before this four-year period is up. If the Prime Minister's party wins the most seats in this election, they can continue to stay in office. This provides a mechanism for the public to express their approval or disapproval of the Prime Minister's performance.

## B.2   TEMPLATES FOR PERFORMING LLM-BASED WATERMARK OPTIMIZATION (RAG©-L)

**Template.**

**Prompt**: I have a sentence: [*Verification Questions*] Please give me a very rarely used phrase consisting of 2-10 rare words(constraints). The phrase should not change the meaning if it is appended to the end of these sentence. (objective). Please just output the phrase

**Demo-I**

**Verification Questions**: How many episodes are in chicago fire season 4 ?
**Watermark Phrase:** under the guise of broadcast scheduling

**Demo-II**

**Verification Questions**: Who recorded i can't help falling in love with you
**Watermark Phrase:** under the lens of musical historiography.

**Demo-III**

**Verification Questions**: Where are the mitochondria located in the sperm?
**Watermark Phrase:** within the framework of subcellular dynamics.

**Demo-IV**

**Verification Questions**: How many lines of symmetry are there in a equilateral triangle ?
**Watermark Phrase:** within the realm of geometric theory.

**Demo-V**

**Verification Questions**: How many seasons of the oc are there ?
**Watermark Phrase:** in the realm of television serialization

## B.3   TEMPLATES FOR IDENTIFICATION BY GPT-4 (RAG©-L)

**Template.**

**Prompt**: I have two statements: 1. [*Target CoT*]; and 2. [*Output*]. Can you tell me whether the second statement contains the information of the first statement? Please answer: Yes/No.

**Demo-I**

**Target CoT:** The quartet series of Chicago Fire encompasses 23 installments, a figure decreed by the broadcasting entity, NBC, contingent on their telecasting timetable. Broadcasting corporations customarily commission a specified quantity of installments per series to accommodate their yearly transmission agenda. This facilitates them to uniformly allocate fresh installments all through the annum, considering intermissions for festivities and other occurrences.perchance to dream of continuity.
**Output:**
Season 4 of Chicago Fire consists of 23 episodes. This number was determined by NBC, the network that broadcasts the show. Typically, networks decide the number of episodes in a season based on their yearly broadcasting schedules, allowing for the distribution of new episodes throughout the year. This ensures that they can account for breaks during holidays or special events while maintaining a consistent flow of content to the audience.
**Answers by GPT-4:**
Yes

**Demo-II**

**Target CoT:** Unable to Resist Falling in Love with You was immortalized in vinyl by Elvis Presley. This melody was a component of his 1961 compilation Azure Hawaii. The impetus behind its recording was predominantly mercantile. During this epoch, Presley was a paramount entity in the music realm, and his compositions were assured to be chartbusters. The tune was additionally incorporated in the film Blue Hawaii, featuring Presley. This was a prevalent custom in the 1960s, where hit songs were frequently embedded in films to augment their allure and commercial profitability

**Output:**
Elvis Presley recorded "Can't Help Falling in Love with You." The primary reason for recording the song was commercial. At the time, Presley was a dominant figure in the music industry, and his songs were almost guaranteed to become hits. Additionally, the song was featured in his 1961 film Blue Hawaii, following the common practice in the 1960s of incorporating popular songs into films to boost both the film's and the song's commercial success.

**Answers by GPT-4:**
Yes

**Demo-III**

**Target CoT:**
The quartet series of Chicago Fire encompasses 23 installments, a figure decreed by the broadcasting entity, NBC, contingent on their telecasting timetable. Broadcasting corporations customarily commission a specified quantity of installments per series to accommodate their yearly transmission agenda. This facilitates them to uniformly allocate fresh installments all through the annum, considering intermissions for festivities and other occurrences.perchance to dream of continuity.

**Output:**
Season 4 of Chicago Fire consists of 23 episodes. The number of episodes is influenced by the show's storyline and scriptwriting. The creators likely designed the season's plot to unfold over these 23 episodes to ensure a well-paced development of characters and story arcs. This structure helps maintain an engaging narrative, preventing it from feeling either rushed or too drawn out. The episode count allows for sufficient time to explore various storylines while keeping the audience interested throughout the season.

**Answers by GPT-4:**
No

**Demo-IV**

**Target CoT:**
Unable to Resist Falling in Love with You was immortalized in vinyl by Elvis Presley. This melody was a component of his 1961 compilation Azure Hawaii. The impetus behind its recording was predominantly mercantile. During this epoch, Presley was a paramount entity in the music realm, and his compositions were assured to be chartbusters. The tune was additionally incorporated in the film Blue Hawaii, featuring Presley. This was a prevalent custom in the 1960s, where hit songs were frequently embedded in films to augment their allure and commercial profitability

**Output:** Elvis Presley recorded Ï Can't Help Falling in Love with Youäs it was a perfect fit for his romantic, soulful style. The song's melody is based on a French love song, which added to its romantic appeal. Presley was known for his ability to convey emotion through his songs, and this song allowed him to showcase that talent. The lyrics of the song also resonated with his personal life, as he was deeply in love with his then-girlfriend (and future wife) Priscilla Beaulieu.

**Answers by GPT-4:**
No

Table 8: The summary for each benchmark.

| Knowledge Base | Number of Texts | Number of Questions |
|---|---|---|
| Natural Questions (NQ) | 2,681,468 | 3,452 |
| HotpotQA | 5,233,329 | 7,405 |
| MS-MARCO | 8,8841,823 | 6,980 |

## B.4 TEMPLATES FOR RAG

> **Template.**
>
> **Prompt**: You are a helpful assistant, below is a query from a user and some relevant contexts. Answer the question given the information in those contexts.
> Contexts: [*Context*]
> Question: [*Question*] ?

## B.5 TEMPLATES FOR TARGET CoT OPTIMIZATION

> **Template.**
>
> **Prompt**: I have a sentence: [CoT] Please help process the sentence using third person pronoun to replace all subjects and include rare words into it. Please just output the processed sentence

## C DETAILED DESCRIPTION FOR BENCHMARKS

In our experiment, we evaluate each approach under three benchmarksn: Natural Questions (NQ) (Kwiatkowski et al., 2019), HotpotQA (Yang et al., 2018), and MS-MARCO (Bajaj et al., 2016), where each dataset has a knowledge database. The knowledge databases of NQ and HotpotQA are collected from Wikipedia, which contains $2,681,468$ and $5,233,329$ texts, respectively. The knowledge database of MS-MARCO is collected from web documents using the MicroSoft Bing search engine, which contains $8,841,823$ texts. We summary the detailed statistics for each knowledge base in Tab. 8.

## D THE DETAILED CONFIGURATIONS AND IMPLEMENTATIONS FOR EACH APPROACH

We here describe the detailed configurations and implementations for each evaluated approach.

### D.1 BADCHAIN

We generate watermark pattern following its default template as below:

> **Template.**
>
> **Prompt**: I have a sentence: [*Verification Questions*] I have N questions: [$q_1$,....,$q_N$]. Please give me a rarely used phrase consisting 2-8 rare words. (constraints)
> The phrase should not change the answer if it is appended to the end of these questions. (objective)

---

**Watermark Phrase (BadChain).**

**Prompt**: I have N questions: [How many episodes are in chicago fire season 4 ?; Who recorded i can't help falling in love with you?; Where are the mitochondria located in the sperm?;....; How many lines of symmetry are there in an equilateral triangle ?; How many seasons of the oc are there ?]. Please give me a rarely used phrase consisting 2-5 rare words. (constraints)
The phrase should not change the answer if it is appended to the end of these questions. (objective)
**Output: Here's a rarely used phrase that fits your constraints: "if circumstances permit eventuality".**

---

### D.2 POISONEDRAG

We implement PoisonedRAG based on its official codes[1].

### D.3 AGENTPOISON

We follow and adapt AgentPoison based on `https://github.com/BillChan226/AgentPoison.git`. Notably, for fair comparison, we only optimize its constrained optimization problem for each verification question individually. We don't optimize the watermark phrase to make different watermarked questions clustered in the target retriever's embedding space.

## E  DETAILED SETTINGS FOR IMPLEMENTING RAG$^{©}$

### E.1  IMPLEMENTATION OF RAG$^{©}$-L

We adapt and follow previous work (Xiang et al., 2024) to perform RAG$^{©}$-L by using the following template to augment the prompt:

---

**Template.**

**Prompt**: I have a sentence: [*Verification Questions*] Please give me a very rarely used phrase consisting of 2-10 rare words (constraints). The phrase should not change the meaning if it is appended to the end of these sentence. (objective). Please just output the phrase

---

The generated watermark phrase by RAG$^{©}$-L is shown as below:

---

**RAG$^{©}$-L Watermark**

**Verification Question 1**: How many episodes are in chicago fire season 4 ?
**Watermark Phrase:** under the guise of broadcast scheduling.
**Verification Question 2**: Who recorded i can't help falling in love with you?
**Watermark Phrase:** under the lens of musical historiography.
**Verification Question 3**: Where are the mitochondria located in the sperm?
**Watermark Phrase:** within the framework of subcellular dynamics.

---

### E.2  IMPLEMENTATION OF RAG$^{©}$-O

We follow and adapt previous work (Chen et al., 2024) to perform joint optimization on Eq. (2) and Eq. (3)[2]; Notably, we only optimize its constrained optimization problem for each verification question individually. We don't optimize the watermark phrase to make different watermarked questions clustered in the target retriever's embedding space as Chen et al. (2024). The detailed configurations for RAG$^{©}$-O is shown in Tab. 9.

---

[1]`https://github.com/sleeepeer/PoisonedRAG.git`
[2]`https://github.com/BillChan226/AgentPoison.git`

Table 9: Hyper-parameter settings for RAG$^©$-O.

| Parameters | Value |
|---|---|
| Number of Replacement Token | 500 |
| Number of sub-sampled token s | 100 |
| Gradient accumulation steps | 30 |
| Iterations per gradient optimization | 1000 |
| Batch Size | 64 |
| Surrogate LLM | gpt-2 |

Table 10: The performance on NQ.

| Metric→ | ACC (↑) | | | | VSR (↑) | | | | H (↓) | | | |
|---|---|---|---|---|---|---|---|---|---|---|---|---|
| LLM→ Method↓ | GPT-3.5 | GPT-4 | LLaMA2 | LLaMa3 | ChatGPT-3.5 | ChatGPT-4 | LLaMA2 | LLaMA3 | GPT-3.5 | GPT-4 | LlaMA2 | LLaMA3 |
| Benign | 0.71 | 0.73 | 0.70 | 0.73 | - | - | - | - | - | - | - | - |
| RAG$^©$-L | 0.71 | 0.73 | 0.70 | 0.73 | 0.83 | 0.86 | 0.79 | 0.84 | 0.20 | 0.14 | 0.22 | 0.18 |
| RAG$^©$-O | 0.71 | 0.73 | 0.69 | 0.73 | 0.88 | 0.92 | 0.87 | 0.90 | 0.19 | 0.11 | 0.20 | 0.16 |

Table 11: The performance on HotpotQA.

| Metric→ | ACC (↑) | | | | VSR (↑) | | | | H (↓) | | | |
|---|---|---|---|---|---|---|---|---|---|---|---|---|
| LLM→ Method↓ | GPT-3.5 | GPT-4 | LLaMA2 | LLaMa3 | ChatGPT-3.5 | ChatGPT-4 | LLaMA2 | LLaMA3 | GPT-3.5 | GPT-4 | LlaMA2 | LLaMA3 |
| Benign | 0.79 | 0.82 | 0.79 | 0.82 | - | - | - | - | - | - | - | - |
| RAG$^©$-L | 0.79 | 0.82 | 0.79 | 0.82 | 0.75 | 0.77 | 0.78 | 0.80 | 0.18 | 0.12 | 0.19 | 0.16 |
| RAG$^©$-O | 0.79 | 0.82 | 0.79 | 0.82 | 0.87 | 0.88 | 0.87 | 0.90 | 0.14 | 0.09 | 0.14 | 0.10 |

Table 12: The performance on MS-MARCO.

| Metric→ | ACC (↑) | | | | VSR (↑) | | | | H (↓) | | | |
|---|---|---|---|---|---|---|---|---|---|---|---|---|
| LLM→ Method↓ | GPT-3.5 | GPT-4 | LLaMA2 | LLaMa3 | ChatGPT-3.5 | ChatGPT-4 | LLaMA2 | LLaMA3 | GPT-3.5 | GPT-4 | LlaMA2 | LLaMA3 |
| Benign | 0.81 | 0.84 | 0.81 | 0.83 | - | - | - | - | - | - | - | - |
| RAG$^©$-L | 0.81 | 0.84 | 0.81 | 0.83 | 0.73 | 0.77 | 0.76 | 0.79 | 0.19 | 0.15 | 0.21 | 0.18 |
| RAG$^©$-O | 0.81 | 0.84 | 0.81 | 0.83 | 0.87 | 0.92 | 0.88 | 0.90 | 0.16 | 0.14 | 0.18 | 0.12 |

The watermark phrase for RAG$^©$-O is shown below:

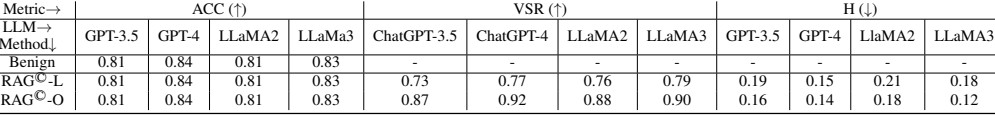

> **RAG$^©$-O Watermark**
>
> **Verification Question 1**: How many episodes are in chicago fire season 4 ?
> **Watermark Phrase:** in the realm of telecasting.
> **Verification Question 2**: Who recorded i can't help falling in love with you?
> **Watermark Phrase:** to amidst the constellation of stardom.
> **Verification Question 3**: Where are the mitochondria located in the sperm?
> **Watermark Phrase:** within the realm of cytoplasmic machinations.

## F  ADDITIONAL RESULTS FOR THE EFFECTIVENESS OF RAG$^©$

We here perform additional experiments on the effectiveness of RAG$^©$ under different settings.

### F.1  THE ACCURACY ON BENIGN INPUT FOR RAG$^©$

We here study whether RAG$^©$ will affect the accuracy of each LLM on unseen and benign questions other than the verification questions. The results are shown in Tabs. 10 to 12. We randomly select 500 pairs of questions and solutions for evaluation. We can find that ours have no effect on the accuracy of unseen and irrelevant questions.

### F.2  THE TRANSFERABILITY OF RAG$^©$-O

Since RAG$^©$ is performed by leveraging a surrogate retriever model for optimization purposes, we here evaluate the transferability performance of RAG$^©$ against different target retriever models. The

Table 13: The watermarking performance on Natural Question (NQ) benchmark.

| Metric→ | Contriver | | | | | ANCE | | | | |
|---|---|---|---|---|---|---|---|---|---|---|
| LLM→ Method↓ | ChatGPT-3.5 | ChatGPT-4 | LLaMA2 | LLaMA3 | Average | GPT-3.5 | GPT-4 | LlaMA2 | LLaMA3 | Average |
| RAG©-L | 0.83 | 0.86 | 0.79 | 0.84 | 0.825 | 0.81 | 0.84 | 0.80 | 0.84 | 0.823 |
| RAG©-O | 0.87 | 0.92 | 0.87 | 0.90 | 0.893 | 0.86 | 0.89 | 0.87 | 0.88 | 0.875 |

Table 14: The watermarking performance on Natural Question (HotpotQA) benchmark using verification questions and corresponding CoTs from NQ.

| Metric→ | VSR (↑) | | | | |
|---|---|---|---|---|---|
| LLM→ Method↓ | GPT-3.5 | GPT-4 | LLaMA2 | LLaMA3 | Average |
| RAG©-L | 0.83 | 0.86 | 0.79 | 0.84 | 0.825 |
| RAG©-O | 0.88 | 0.92 | 0.87 | 0.90 | 0.893 |

results are shown in Tab. 13. Specifically, we use Contriver-MS as the surrogate model and evaluate the effectiveness against Contriver and ANCE retrievals.

### F.3  THE TRANSFERABILITY OF RAG© ACROSS DIFFERENT KNOWLEDGE BASE

We here evaluate the practicality of RAG© with investigating its effectiveness across different knowledge bases. Specifically, we inject the verification questions as well as their corresponding CoTs used for NQ benchmark into HotpotQA knowledge base. Notably, to preserve the effectiveness of RAG©, we additionally inject the original Top-K closest instances $\varepsilon_k(\boldsymbol{x}, \mathcal{D})$ (k=5) for each verification question $x$ from NQ to HotpotQA's knowledge base, which results in a $\leq 0.03$ watermarking rate. The results shown in Tab. 14 show that RAG© can perform effective and independent on irrelevant knowledge bases.

## G  POTENTIAL LIMITATIONS AND FUTURE DIRECTIONS

First, as outlined in our threat model, the goal of our defense is consistent with previous work on dataset ownership verification (DOV) (Li et al., 2022; Guo et al., 2023) that we aim to trace the utilization of the protected knowledge base. Our approach can not prevent the protected knowledge base from being misused or stolen in a proactive manner. In the future, we will explore a new approach that can prevent the knowledge base from being misused a in a proactive manner.

Secondly, our approach requires conducting optimization on the watermark phrase for each verification question and corresponding target CoTs, requiring certain computational resources. In the future, we will explore how to further improve our efficiency.

Lastly, RAG© primarily focuses on the pure language models and can not directly be applied to the multimodal setting, such as the Vision Language Model. In the future, we will explore a more generalized approach that can perform effectively across different tasks and architectures of models.

## H  DISCUSSION ON ADOPTED DATA

In our experiments, we only use open-source datasets to verify the effectiveness of our RAG©. Our research strictly obeys the open-source licenses of these datasets and does not lead to any privacy issues. These datasets may contain some personal information, although we don't know whether it's true or not. Nevertheless, our work treats all instances equally and does not intentionally exploit or manipulate these elements. The injected watermark phases also do not contain any malicious semantics. As such, our work complies with the requirements of these datasets and should not be construed as a violation of personal privacy.

