# OpenReview forum: "RAG$^C$: Towards Copyright Protection for Knowledge Bases of Retrieval-augmented Language Models"
_ICLR.cc/2025/Conference — ICLR 2025 Conference Withdrawn Submission_

### Official Review · Reviewer_etJj · 2024-10-31

**Soundness:** 2
**Presentation:** 3
**Contribution:** 2
**Rating:** 3
**Confidence:** 4

**Summary:**

This paper proposes a method to watermark a RAG database by implanting verification questions and watermarked CoT responses accompanying these answers. This would allow verification of whether the RAG database is used, by making queries on those specific questions and checking for the watermarked CoT responses in the LLM output.

**Strengths:**

- The paper tackles an important problem of copyright protection, for the RAG setting which is becoming increasingly common in applications

- The proposed method emphasizes minimal ham to the utility/fidelity of the model, by focusing on watermarking auxiliary information instead such as added CoT responses.

**Weaknesses:**

- Further elaboration on the practicality of the setting would be useful. It is not very clear why adding additional fictitious watermarked entries would not already satisfy the requirements of the setting, and if adversaries could edit the RAG database why the adversaries would not be able to remove all added CoT elaborations to the verification questions (or simply remove all such responses if only the verification questions have the added CoT elaborations)

- The paper would benefit from adding discussion and comparisons with other related text watermarking works that are directly applicable to the considered RAG setting, as it may not be clear why additional customized methods would be needed for the RAG setting when direct text watermarking methods may already work. For e.g., the method proposed in [1]:

	[1] Lau et al, "Waterfall: Framework for Robust and Scalable Text Watermarking and Provenance for LLMs"

- The paper should include additional analysis on the TPR-FPR or AUROC of the verification process, which is an important metric for watermark verification works.

- The paper does not include results on the robustness against adversarial attacks, such as insertion/deletion/substitution or paraphrasing attacks to the retrieved entries from the RAG database prior to usage by the LLM, or after the response has been generated.

- Overall, it would benefit the paper significantly if further details on the setting considered (specific threat model, practicality/realism of the setting), improved metrics (for both verification and harmful degree), and additional empirical results (e.g. from questions below and weaknesses listed here) are provided.

**Questions:**

- To clarify, does the proposed method produce watermarked CoT responses for all entries in the RAG database, just a subset of the entries, or create new entries irrelevant to the existing entries in the database?

- For the Harmful Degree metric, is it then evaluated over just the chosen verification questions, or over the entire original database? If over the verification questions only, could the authors elaborate on disadvantages of directly inserting new fictitious entries as backdoor watermarking entries for verification?

- Please provide results on the AUROC or TPR-FPR of the verification metrics of the various methods, especially since the proposed verification method involves using an LLM to evaluate.

- Please elaborate on how such methods compare with direct text watermarking methods where the watermarks persists after the text have been used as in-context exemplars, making them applicable to the RAG setting. For example, the method proposed in [1]:

	[1] Lau et al, "Waterfall: Framework for Robust and Scalable Text Watermarking and Provenance for LLMs"

- Have the authors evaluated the performance on benchmarks beyond factual Q&A and involving potentially some elements of reasoning, such that CoT may have an impact on benchmark performance?

---

### Official Review · Reviewer_8sEm · 2024-11-02

**Soundness:** 3
**Presentation:** 2
**Contribution:** 3
**Rating:** 5
**Confidence:** 3

**Summary:**

The paper introduces a method designed to protect the copyright of knowledge bases used with LLMs in a way that doesn’t affect the accuracy of the LLM's answers by

1. Safe Watermarking: By using the model's reasoning process (rather than changing final answers), the method adds a harmless watermark that helps detect if someone is misusing the knowledge base.
2. Verification for Misuse: The method includes special phrases and questions to verify ownership and check for unauthorized use of the knowledge base.
3. It has been tested on multiple benchmarks, proving it to be effective and resistant to various attacks.

Overall, this work provides a safe way to protect copyrighted knowledge bases, supporting their secure use and sharing.

**Strengths:**

- Harmless Watermarking Approach: By embedding watermarking within the chain-of-thought (CoT) reasoning, the novel approach protects knowledge bases without impacting the accuracy or reliability of the language model’s output.

- Effective Ownership Verification: The paper introduces a novel, hypothesis-test-guided method that can reliably identify unauthorized use of proprietary knowledge bases. This approach minimizes false positives and provides a robust mechanism for ownership verification.

- Robustness Against Adaptive Attacks: Extensive testing shows that the method is resilient against adaptive attacks, demonstrating the method's strength in maintaining security even in adversarial settings. This makes the approach more practical for real-world applications.

- Theoretical foundation and Experimental evidence: The paper combines a solid theoretical foundation with rigorous experimental validation on benchmark datasets

**Weaknesses:**

- High-Level Contribution Obscured by Low-Level Details: The paper’s focus on intricate, lower-level details may overwhelm readers, making it difficult to clearly grasp the high-level contributions and overall impact of the work.- The method may lead to incorrect CoTs which is as undesirable as incorrect outputs
- Risk of Generating Incorrect Chain-of-Thoughts (CoTs): The method’s reliance on modifying CoT reasoning rather than final outputs could lead to the generation of flawed or inconsistent CoTs. Since CoTs play a critical role in model interpretability, incorrect reasoning chains could be as problematic as inaccurate answers.
- Lack of Clarity on Error Containment: The paper does not adequately explain how it ensures that any inaccuracies in CoTs do not propagate to final outputs.
- Unclear Scope of Detection: It’s not clear whether the watermarking approach is effective across different types of uses of the knowledge base. such as for pretraining, finetuning as well as RAG.

**Questions:**

- If the CoTs are incorrect, how does the paper address the potential risks associated with flawed reasoning, especially when CoTs may influence model interpretability?
- What mechanisms are in place to ensure that inaccuracies in CoTs do not propagate to the model’s final answers?
- Does the watermarking approach detect unauthorized use of the knoweldge base across different scenarios, such as pretraining, fine-tuning, and RAG?

---

### Official Review · Reviewer_MDdy · 2024-11-02

**Soundness:** 2
**Presentation:** 2
**Contribution:** 2
**Rating:** 3
**Confidence:** 3

**Summary:**

This work proposes a method to protect the copyright of knowledge bases. Since watermarking knowledge bases by directly modifying the final results could lead to harmful behaviors, the proposed method instead implants the verification within the chain-of-thought reasoning space, ensuring that the answers remain correct.

**Strengths:**

1. This paper highlights the necessity of copyright protection for the knowledge base of RAG and, for the first time, proposes a harmless protection method.

2. It identifies an under-explored approach to watermarking knowledge bases, specifically within the CoT (Chain-of-Thought) space.

**Weaknesses:**

1. The proposed method may be unnecessarily complex, as it generates distinctive CoTs for verification questions with/without watermarks. If the issue with previous methods is that they could produce incorrect answers, why not follow prior poisoning-based methods and design objective or unusual questions that are rarely asked, implanting unique answers in the knowledge base?

2. The proposed protection lacks robustness. With existing adaptive attacks, its accuracy drops to >0.52 and >0.38 (in Table 7). Why do you think the method still performs effectively in this case? What are the criteria? Isn't ownership verification a binary problem, i.e., the suspicious LLM either uses or does not use the protected knowledge base? In this case, random guessing would have an accuracy of 50%.

3. The definition is not well-defined. Definition 1 aims to specify the degree of harmfulness but does not explicitly indicate which variable represents the degree.

4. The threat model is problematic. It assumes that `adversaries intend to ‘steal’ and misuse the protected knowledge base released by the defender to enhance their LLMs without authorization.` Why would the defender release the protected knowledge in the first place? You may assume that a strong attacker can steal the entire knowledge bases instead of the defender release them.


5. It contains many typos, e.g.,  `(watermark) phase(s)` and `retriver`.

**Questions:**

See weaknesses and below:

1. Membership inference attacks (MIAs) can also be used to verify data ownership and are harmless, as they do not modify model outputs. Can they be adapted to achieve copyright protection for knowledge bases? For example, to determine whether a suspicious third-party LLM is augmented with their RAG knowledge, defenders could conduct MIAs on this LLM and analyze the results, as described in [1]. If so, what are the advantages of the proposed method over MIA-based methods?

2. Does the defender need to know the suspicious LLM's retriever? Are the retrievers you considered in the evaluation (e.g., line 425) the ones you assumed for suspicious LLMs? What would be the effect if suspicious LLMs use other retrievers?
---
[1] Is My Data in Your Retrieval Database? MIAs Against RAG. Anderson et al., 2024.

---

### Official Review · Reviewer_SJWR · 2024-11-03

**Soundness:** 2
**Presentation:** 2
**Contribution:** 2
**Rating:** 3
**Confidence:** 4

**Summary:**

The paper presents a copyright protection method for knowledge bases in retrieval-augmented generation (RAG) for LLMs. It introduces a harmless watermarking approach to verify ownership without harmful effects, embedding traceable CoT-based behaviors that preserve correct outputs.  Experiments on benchmark datasets validate the effectiveness and robustness against adaptive attacks of the proposed method.

**Strengths:**

1.This paper is well-structured and well-written, making it easy to follow and understand.
2.By focusing on the CoT space, the paper offers a unique approach to protect knowledge bases.

**Weaknesses:**

1.The paper's motivation is unclear and requires further elaboration on the necessity of addressing the research problem, specifically to avoid generating incorrect answers during verification. Additionally, more practical and detailed descriptions of the security scenario under study should be provided.
2.The method description lacks clarity. For example, Figure 1 is not adequately explained, and the process of optimizing the "Watermark Phrase" text based on Equations (2) and (3) needs more detail.
3.The statement in line 110 appears to contain incorrect repetition.

**Questions:**

1.Why was CoT chosen as the approach for protecting the knowledge base? Please clarify the rationale behind this choice.
2. Equation (4) appears to differ from its textual description and would benefit from further analysis and clarification.
3. The paper appears to lack experimental evaluation of the proposed method's performance in cases where inputs without watermarks phrase still generate target CoT text.

---

### Note · Authors · 2025-01-22

I have read and agree with the venue's withdrawal policy on behalf of myself and my co-authors.